

# Preparation of low concentration $H_2$ test gas mixtures in ambient air for calibration of $H_2$ sensors.

Niklas Karbach[1], Lisa Höhler[1], Peter Hoor[2], Heiko Bozem[2], Nicole Bobrowski[3,4], Thorsten Hoffmann[1*]

[1] Department of Chemistry, Johannes Gutenberg-University Mainz, 55128 Mainz, Germany
[2] Institute for Atmospheric Physics, Johannes Gutenberg-University Mainz, 55128 Mainz, Germany
[3] Institute of Environmental Physics, University of Heidelberg, 69120 Heidelberg, Germany
[4] Istituto Nazionale di Geofisica e Vulcanologia (INGV), Osservatorio Etneo, 95125 Catania, Italy

*Correspondence to*: t.hoffmann@uni-mainz.de

**Abstract.** Using electrochemical gas sensors for quantitative measurements of trace gas components in ambient air introduces several challenges, of which interference, drift and aging of the sensor are the most significant. Frequent and precise calibration as well as thorough characterization of the sensor helps to achieve reliable and repeatable results. We therefore propose the use of a simple, lightweight and inexpensive setup to produce hydrogen calibration gases with precisely known concentrations in ambient air. The hydrogen is produced by electrolysis with electric current monitoring and the output can be set to any value between ~3 $\mu g_{H2}$/min and ~11 $\mu g_{H2}$/min. With a dilution flow of 500 mL/min, for example, this results in a concentration range from ~70 ppm up to ~240 ppm, but concentrations significantly below or above this range can also be covered with accordingly modified dilution flows. This setup can be used not only for calibration, but also for thorough and long-term characterization of electrochemical gas sensors to evaluate sensitivity, zero voltage and response time over extended periods of time.

## 1    Introduction

A major challenge for electrochemical measurements of trace gas components in ambient air are potential errors caused by interference, drift and aging of gas sensor electrodes (Pang et al., 2017; Roberts et al., 2017; Roberts et al., 2012; Jasinski et al., 2018; Aiuppa et al., 2011). Frequent calibration and thorough characterization of the sensors used are key to measure correct concentrations and thus to generate high-quality data (Kamionka et al., 2006; Hasenfratz et al., 2012; Tian et al., 2019; Korotcenkov et al., 2009). The environmental conditions at the measurement site can significantly change the response and sensitivity of the sensors (matrix effects) (Baron and Saffell, 2017; Pang et al., 2017; Lewis et al., 2016; Jasinski et al., 2018; Tian et al., 2019; Korotcenkov et al., 2009; Farquhar et al., 2021). The quality of measurements carried out with electrochemical sensors can therefore be significantly improved if the sensors are calibrated under ambient conditions directly before the measurement. Known varying influences on the measurement signal, such as humidity or temperature, can be partially corrected by characterization of the sensors beforehand and therefore the possibility to compensate for those influences afterwards. However, unknown perturbations can cause systematic errors that are difficult to detect and often lead to misinterpretation of the experimental data (Lewis et al., 2016; Roberts et al., 2017; Pang et al., 2017). More recently,



artificial neural networks and machine learning methods have been used to convert the raw data into reliable concentration values, taking into account environmental parameters such as temperature, humidity and data from other gas sensors in the measurement environment (Wei et al., 2018; Cross et al., 2017; Zimmerman et al., 2018).

Standard gas mixtures in pressurized containers are an obvious option for regular calibration. However, the storage of hydrogen
standard gas mixtures over longer periods of time can lead to changes in concentration, as was investigated for steel and aluminium canisters with hydrogen diluted with ambient air (Jordan and Steinberg, 2011). They found that the hydrogen concentration in aluminium canisters in particular can increase significantly in the first few months of storage. Storing gas samples and calibration mixtures in plastic bags (e.g. Tedlar bags) leads to even greater changes in concentration when stored for days to a few weeks (Schuette, 1967) (Barratt, 1981). Especially for hydrogen, with its exceptionally high diffusion and
permeation coefficient, plastic bags are not suitable over longer periods of time. Therefore, a method for the rapid, efficient and accurate production of hydrogen as a calibration gas at low cost is required. Electrolysis offers a simple method of producing high purity hydrogen with the ability to control the amount of hydrogen produced by controlling the electric current flowing through the electrolysis cell. Continuous production of hydrogen enables long-term performance evaluation (drift, sensitivity, zero voltage, response time) of sensors and pulse width modulation (PWM) enables automatic calibration.

The purpose of this paper is to present a method for producing $H_2$ test gas mixtures that has been specially developed for the calibration of $H_2$ sensors. In particular, the ability to take a simple and robust calibration system into the field distinguishes the system presented here from existing technologies.

## 1.1 Electrolysis

In recent years, hydrogen has gained popularity as a form of chemical energy storage and as a substitute for fossil fuels, as it
can be easily produced by electrolysis and converted back into electricity via fuel cells in times of high energy demand (Zhang et al., 2016; Tarhan and Çil, 2021; Wang et al., 2014). In a typical electrolysis cell, water is split into hydrogen and oxygen at a theoretical decomposition voltage of 1.23 V. However, due to overvoltage, the actual voltage required is much higher and the efficiency of electrolysis decreases (Carmo et al., 2013).

However, according to Faraday's first law of electrolysis, the amount of hydrogen produced depends only on the electric current
flowing through the cell (Zeng and Zhang, 2010). Therefore, monitoring the electric current provides a measure of the amount of hydrogen being produced. By controlling the electric current, e.g. by PWM or adjusting the applied voltage, the amount of hydrogen produced can be controlled.

The current yield of the system describes the yield of the electrolysis by comparing the product quantity with the product quantity predicted according to Faraday's first law of electrolysis. A current efficiency of 100 % describes a "perfect" cell and
allows the exact amount of hydrogen produced to be calculated by measuring the electric current flowing through the cell.





## 1.2 Calibration of electrochemical H₂ sensors

Commercially available calibration gas mixtures are usually produced in pure (dry) nitrogen, which leads to considerable variations in the sensor response when these mixtures are used for the direct calibration of electrochemical sensors, which are then used for measurements in ambient air. Especially for electrochemical hydrogen sensors, the water content of the sampled gas is the most important influencing factor, not only because humidity is a highly variable component depending on the sampling location, but also because water molecules can influence the electrode surfaces or electrolyte concentrations, and not least because water is a product of the reaction of the hydrogen measurement within the electrochemical cell itself ($2H_2 + 2O_2^- \rightarrow 2H_2O + 4e^-$). A higher level of control offers the use of artificially humidified zero air or ambient air for mixing with hydrogen or standard calibration gas mixtures. There are two options here: either dynamic production, where mass flow controllers (MFCs) are used to dilute pure hydrogen or hydrogen/nitrogen mixtures with ambient air/humidified zero air (Benammar et al., 2020; Domansky et al., 1998), or batchwise production of calibration gases in e.g. Tedlar bags or pressurized cylinders (Korotcenkov et al., 2009; Karbach et al., 2022; Rüdiger et al., 2018). The first option is more demanding in terms of instrumentation, but enables continuous measurement, while the second option is faster and cheaper, but only allows for batched production of test gases. Another commonly used method is the calibration of the sensor by simultaneous measurement with a validated reference device (Malings et al., 2019).

In addition, for the controlled dynamic production the use of electrolysis with electric current monitoring could be applied, which allows for precisely known amounts of hydrogen being produced on demand as well as eliminating the need for a pressurized cylinder of hydrogen or MFCs in the laboratory, and therefore significantly reducing the cost and size of calibration equipment.

## 2 Materials and Methods

The setup consists of an electrolysis cell with 9 % acetic acid (AcOH) in deionized water as the electrolyte, a stainless-steel cathode and platinum wire anode. A controllable lab bench power supply provides power. A high resistance (100 Ω) shunt resistor in series with the electrolysis cell allows for an accurate current measurement with a microcontroller and appropriate ADC. The hydrogen produced is discharged via connected tubes that lead to a T-piece, where it is mixed with the dilution air (fresh ambient air, no hydrogen added) and directed to the sensor to be calibrated. The combination of electrodes reduces the



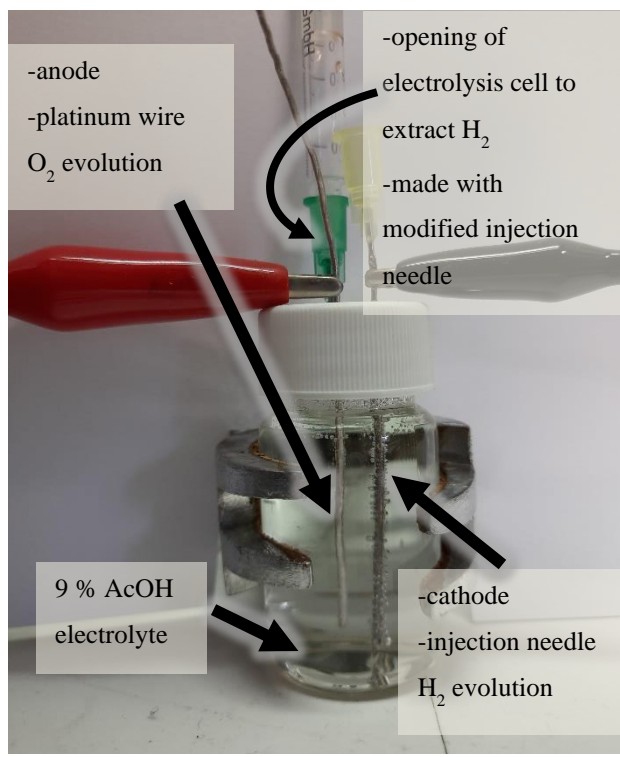

- anode
- platinum wire
$O_2$ evolution

- opening of electrolysis cell to extract $H_2$
- made with modified injection needle

9 % AcOH electrolyte

- cathode
- injection needle
$H_2$ evolution

**Figure 1: Photograph of the experimental setup of the electrolysis cell.**

cost of the setup drastically, as only one platinum electrode is needed. In all experiments, the hydrogen was diluted with a constant flow of 500 ml/min of fresh ambient air (note: ambient air has a slight background of 0.5 ppm $H_2$, however, this is well below the ranges that were tested in this study). The hydrogen concentration was changed by adjusting the electric current flowing through the electrolysis cell. To measure the hydrogen concentration that is created by the system, an electrochemical "Alphasense H2-BF" sensor is used, that was calibrated by creating gas mixtures of differing concentrations by diluting pure hydrogen with the appropriate amount of fresh ambient air. These sensor readings were then used to calculate the current yield of the system.

To further reduce costs, a setup consisting of two stainless steel electrodes was also tested. With this setup, the current yield stabilized after ~12 hours, but was well below 100 %, which prevented its use as a primary calibration standard.


# 3    Results

Figure 2 shows both the measured $H_2$ concentration and the theoretical $H_2$ concentration calculated using Faraday´s first law of electrolysis, and the corresponding current yield. The sensor was calibrated with calibration gases prepared by introducing

precisely known amounts of hydrogen into a known volume of ambient air in a Tedlar bag. This gas mixture was measured directly after preparation to avoid changes in the concentration of the calibration mixture. The calibration yielded a good coefficient of determination ($R^2$) and was reproducible. The raw data for a calibration run can be seen in Fig. 3. Figure 3 contains the calibration equation with the coefficient of determination of the calibration.

As can be seen in Fig. 2, after changing the electric current flowing through the electrolysis cell, the concentration at the outlet

did not change instantaneous (like the measured electric current), but instead needed time to adjust to the correct value. This limits the number of concentration changes that this system can produce in a certain amount of time. Repeatability tests have been conducted with new cathodes and fresh electrolyte. The tests revealed that after an initial run-in phase of about 2 hours,



the system always approaches a current yield of 1.0, which expresses that the H₂ output that is predicted by Faraday´s first law of electrolysis and the measured H₂ output are exactly the same. This shows that the true hydrogen output can consistently be

derived from the measured current, allowing to accurately produce hydrogen standard mixtures. A more detailed description of repeatability experiments is given in the supplemental information.

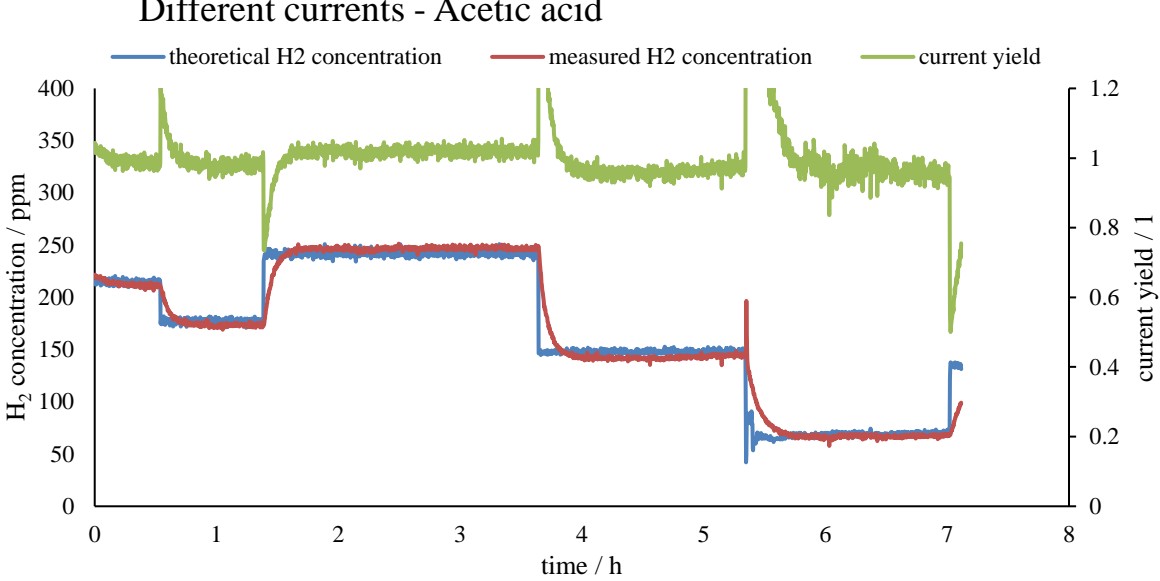

**Figure 2: Plot of the measured H₂ concentration in orange (as calibrated with the external calibration shown in Fig. 3). The theoretical H₂ concentration as calculated with Faraday´s first law of electrolysis in blue, and the current yield in green.**


**Table 1: Raw data of the measured electric current flowing through the electrolysis cell, measured hydrogen concentration (as calibrated with the external calibration) and corresponding current yield. The respective errors are one standard deviation of the measured data. The error of the theoretical concentration is given by the error of the current measurement.**

| time / h | current / mA | theoretical conc. | measured conc. | current yield / 1 |
|---|---|---|---|---|
| 0  -  0.51 | 14.06 ± 0.05 | 214.3 ppm | 213 ± 3 ppm | 1.00 ± 0.02 |
| -  1.35 | 11.61 ± 0.03 | 177.0 ppm | 176 ± 5 ppm | 0.99 ± 0.03 |
| -  3.59 | 15.88 ± 0.04 | 242.0 ppm | 245 ± 6 ppm | 1.01 ± 0.03 |
| -  5.31 | 9.70 ± 0.03 | 147.8 ppm | 145 ± 9 ppm | 0.98 ± 0.06 |
| -  6.97 | 4.51 ± 0.11 | 68.6 ppm | 69 ± 6 ppm | 1.01 ± 0.11 |





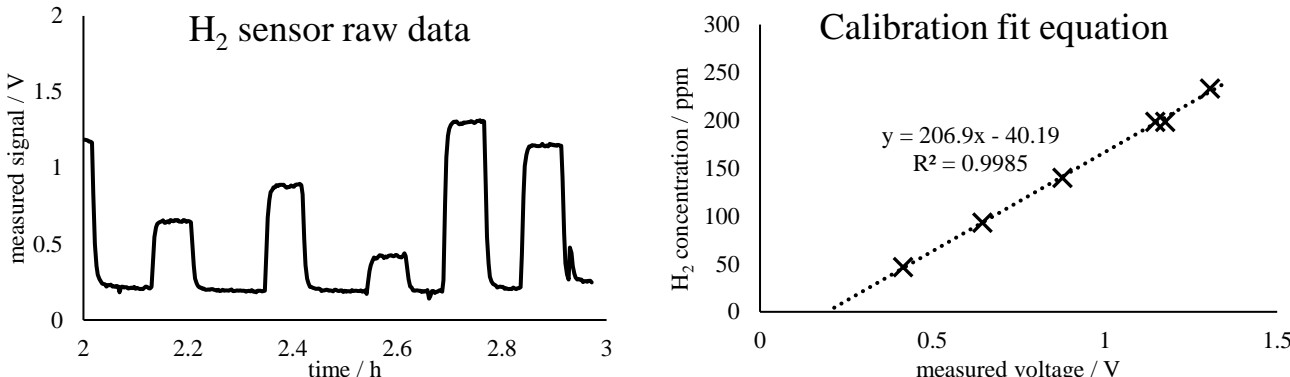

**Figure 3: (left) time series of the measured voltages (raw data) during calibration.    (right) Plot of the averaged voltages measured during the corresponding calibration plotted against the concentration of the calibration gas mixture. The linear fit equation, as well as the coefficient of determination is given in the plot.**

## 4    Discussion

Figure 2 and Table 1 show that the current yield is close to unity over the entire concentration range tested. This shows that the actual performance of the electrolytic cell is close to what is predicted by Faraday's first law of electrolysis. Thus, this type of calibration setup can be used not only to produce hydrogen gas mixtures at low concentrations over several days, but also to directly calculate the concentration of the gas mixture by measuring the current and applying Faraday's first law of electrolysis. This setup can therefore be used as a primary standard for the calibration of hydrogen sensors.

The production of calibration gases with precisely known concentrations is also possible by other means, including those used in this work for the reference calibration of the $H_2$ sensor. However, these methods are prone to errors caused by inaccurate or incorrect working procedures and only allow the calibration gases to be produced in batches. In addition, the high diffusion coefficient of hydrogen makes it necessary to prepare the calibration gas immediately before the measurement to avoid the loss of hydrogen by diffusion through the Tedlar bag. If a continuous supply of calibration gas is required, it is usually necessary to use mass flow controllers to dilute pure hydrogen with ambient air. However, this involves considerable costs and equipment (MFCs, gas cylinders, pumps, etc.). Using an electrolysis cell with accurate monitoring of the electrical current reduces the instrumentation to a low-cost power supply, a microcontroller, some resistors, and the electrolysis cell itself, which consists of a platinum wire anode and a stainless-steel cathode immersed in ~9 % acetic acid. The hydrogen produced can then be fed into the gas flow using a simple T-piece.

The calibration setup presented in this paper is comparatively inexpensive, but still of similar quality to manual batch calibration with Tedlar bags. The complete setup is extremely lightweight (~ 300 g with battery as the power source), so that a mobile calibration station is possible. Such a mobile calibration station would allow for calibrating sensors directly in the field before measurement, using ambient air as a dilution medium. This would significantly reduce matrix effects as changes

in ambient parameters (T, p, RH, concentration of other trace gases, ...) between calibration and measurement would be minimized.

Other gases like $Cl_2$, CO, $H_2S$, $N_2$, NO, $O_2$, $O_3$, $AsH_3$ and $SbH_3$ (see review of (Barratt, 1981) and (Hsu et al., 2015)) may also be produced via electrolysis and therefore be used for the calibration of gas sensors and the production of gas mixtures with precisely known concentrations. However, it is strongly advisable to determine the current yield of the specific setup (chemicals, electrodes, applied voltage, electrolyte, …) prior to calibration to avoid systematic errors of the measurements due to wrong calibration. A correction factor can then be used to account for non-ideality of the electrolysis.

**5    Conclusions**

In summary, the system presented here can accurately generate and reproduce a stable flow of gas mixtures of known concentrations over several days using ambient air as a dilution medium. In combination with the small size and low weight of the system, this enables the calibration of hydrogen sensors in the field, reducing the influence of matrix effects on the accuracy of the sensor. The system is inexpensive to assemble and easy to maintain, allowing frequent calibration without

much manual effort, which is the key to reliable measurement results. The design could also be further improved with a fixed voltage source (e.g. a battery) in combination with pulse width modulation (PWM), with the aim of being able to control the current flowing through the electrolytic cell more easily. This would further reduce the size and weight of the system and allow automatic calibration of the sensors by adjusting the current flow after a predefined period of time.

**Author Contributions:** NK and TH conceived and designed the experiments; NK and LH performed the experiments; NK, LH and TH analyzed the data; all authors participated in writing the manuscript.

**Conflicts of Interest:** The authors declare that they have no conflict of interest.

**Acknowledgments:** The authors thank Siegfried R. Waldvogel (Johannes Gutenberg-University Mainz) for supplying the platinum wire to conduct the experiments.
**Funding:** This research was funded by the Deutsche Forschungsgemeinschaft (DFG, German Research Foundation) – TRR 301 – Project-ID 428312742 – and DFG Project HO 1748/23-1 – and from TeMaS (Terrestrial Magmatic Systems, a collaborative effort of the Universities of Mainz, Frankfurt and Heidelberg).

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
