# Peer review of "Preparation of low concentration H2 test gas mixtures in ambient air for calibration of H2 sensors."

_EGUsphere, 2024_

## Author Comment (AC1)

Dear Dr. Duclaux,

Thank you for your helpful comments, which we have used to improve the clarity and readability of the manuscript. In the following, we address the individual comments point by point. The reviewers' comments are shown in **blue**, our response is in **black** and we show additions/changes to the manuscript in **red**.

**RC2**

Dear authors, thank for precedent clarification.

Based on the SI, the long term test confirms the repeatability of the method. Concerning stability over time, the current yield stay close to 1 during 14 hours. Anyway hydrogen production seems stable during the first two hours after the initial run-in phase, then a regular drift is observed (from 2 to 14 h, with an increase of +20 ppm). We notice "peaks" at regular intervals as if compensation has been made.

Could you explain what experimentally leads to this ?

Best regards.

**Concerning the "drift"**

You are absolutely correct that the amount of hydrogen increases over the duration of the experiment. This drift is real and not a measurement artifact. Below we give reasoning for why this drift is not problematic for the application of the system.

Initially we were unsure if to include your mentioned graphic, as we expected this question. However, we decided to include this graphic as this plot nicely shows that you only have to measure the electric current to accurately predict the amount of hydrogen that is produced. If the current increases over the duration of the experiment, the amount of hydrogen also increases in the same proportion. This makes the output amount of hydrogen produced by the system perfectly predictable by only measuring the electric current flowing through the cell.

One might argue if the current increased as a consequence of increasing surface roughness of the electrodes (leading to a larger surface area -> lower resistance), or simply by voltage drifts of our (low cost) power supply. However, in both cases the results are the same. The $H_2$ concentration increases in the same proportion as the current increases. This is shown by the constant current yield of ~1, which describes the measured $H_2$ concentration divided by the theoretical $H_2$ concentration as is predicted by the measured current.

**Concerning the "peaks"**

The experiment was started in the evening and was running over night inside an otherwise unused fume hood in an air-conditioned lab. No people were present during the whole duration of the experiment, so influences / interferences from other experiments are out of question. From previous experience with electrochemical sensors, it is known that EC sensors are sensitive towards changes in both RH and T. Depending on the specific type of sensor that is used, when e.g. humidity changes, the sensor needs time to adjust to the new RH. This adjustment can be seen in the measurement signal in a form very similar to the form that can be seen in the mentioned plot. We therefore conclude that the peaks you can observe are likely related to artifacts from air conditioning. However, the sensor adjusts in the matter of 10 to 20 minutes to the new condition, so this behavior has no influence on the results itself.

We have also clarified and explained this behavior in the SI.

Thank you for the constructive question/remarks!

"The amount of hydrogen increases over the duration of the experiment. However, this drift has no influence on the predictability of the system itself, as the electric current increases in the same proportion as the measured (real) hydrogen concentration. Therefore, the current yield remains at ~1 and the hydrogen output stays absolutely predictable by measuring the electric current and applying Faraday´s law of electrolysis. The reason, why the electric current increases might be explained by an increasing surface roughness (and therefore a larger surface area which leads to a lower electric resistance of the electrolysis cell), or voltage drifting of our (low cost) power supply.

The "peaks" that are visible in Fig. 1 are likely due to changes in relative humidity. Out of question are influences by other experiments, or people residing in the lab, as the experiment was conducted over night. From our experience, we can say that electrochemical sensors are sensitive to humidity changes and a sudden change (as might be caused by switching of the air conditioning in the lab) causes the sensor to behave in the way as is happening in the experiment series shown in Fig. 1. As the sensor adjusts in a matter of 10 to 20 minutes to the new humidity, this behaviour has no influence on the results itself."

---

## Author Comment (AC2)

Dear Dr. Etiope,

Thank you for your helpful comments, which we have used to improve the clarity and readability of the manuscript. In the following, we address the individual comments point by point. The reviewers' comments are shown in **blue**, our response is in **black** and we show additions/changes to the manuscript in **red**.

**RC1**

I would like to suggest the following:

- better clarify (also in Abstract and Conclusions) the range of concentrations of H2 for which the proposed system is valid. I see a range of 50-250 ppmv in the tests;

We have tested the system in the range of 69 to 242 ppmv. For those measurements, we have chosen a flowrate of the dilution air of 500 mL/min which we kept constant for all experiments. The production rate of hydrogen therefore corresponds to 3.07 $\mu g_{H2}$/min to 10.75 $\mu g_{H2}$/min. We think it is better to give a range of valid production rates, rather than a range of valid concentrations, as by adjusting the dilution factor (i.e. the flowrate of the pump), the concentration can be as high or as low as is practically possible.

For the initial quick review, we have already clarified this in the manuscript:

"The hydrogen is produced by electrolysis with electric current monitoring and the output can be set to any value between ~3 $\mu g_{H2}$/min and ~11 $\mu g_{H2}$/min. With a dilution flow of 500 mL/min, for example, this results in a concentration range from ~70 ppm up to ~240 ppm, but concentrations significantly below or above this range can also be covered with accordingly modified dilution flows."

- indicate related applications for that range (I assume not for atmospheric measurements).

The purpose of the system is to easily produce calibration gas mixtures with precisely known concentrations that can then be used to calibrate and characterize various types of hydrogen sensors. As the concentration of the produced calibration gas mixture can nearly arbitrarily be adjusted by adjusting the dilution factor, possible applications are broad.

By choosing a high dilution factor, a low concentration of the calibration gas mixture can be achieved, which allows to calibrate highly sensitive sensors (environmental measurements, finding low concentration hydrogen sources, ...). On the other hand, choosing a lower dilution

factor leads to higher concentrations, which allow to calibrate less sensitive sensors (industrial applications, reaction monitoring, ...).

- be more rigorous in the use of the terms "sensitivity" (which is voltage vs concentration) and "accuracy" (measured concentration vs real concentration), and to report (Abstract and Conclusion) the accuracy of the standard that can be obtained.

The accuracy of the standard that is obtained is directly proportional to the current yield of the reaction, that is shown in the diagrams. A current yield of 1.00 means that the measured concentration is the same concentration as is predicted by applying Faraday´s first law of electrolysis and incorporating the dilution factor (500 mL/min in our experiments). The current yield calculates according to the following formula:

$$\eta_c = \frac{c_{\text{measured}}}{c_{\text{theoretical}}}$$

A current yield of 1.00 results in a totally accurate standard, as the concentration is calculated according to Faraday´s law.

The description can be found in the supporting information of this manuscript.